# Comprehensive Review of Acute Pancreatitis Pain Syndrome

**Jacob Beiriger, Adnan Khan \*, Brian Yan, Heather Ross, Makala Wang, Michael Carducci** [ID]**, Natalia Salinas Parra** [ID]**, Salil Chowdhury, Ryan Erwin, Paul Forrest, Sarah Chen and Alexis Gerber**

Department of Internal Medicine, Thomas Jefferson University Hospital, Philadelphia, PA 19107, USA; jacob.beiriger@students.jefferson.edu (J.B.); brian.yan@students.jefferson.edu (B.Y.); heather.ross@students.jefferson.edu (H.R.); makala.wang@students.jefferson.edu (M.W.); michael.carducci@students.jefferson.edu (M.C.); natalia.salinas-parra@students.jefferson.edu (N.S.P.); salil.chowdhury@jefferson.edu (S.C.); ryan.erwin@jefferson.edu (R.E.); paul.forrest@students.jefferson.edu (P.F.); sarah.chen@students.jefferson.edu (S.C.); alexis.gerber@jefferson.edu (A.G.)
\* Correspondence: adnan.khan@jefferson.edu

**Abstract:** Pancreatitis is a condition that causes inflammation in the pancreas, an organ located behind the stomach. This condition often presents as neuropathic, inflammatory, and/or visceral pain. Acute pancreatitis is typically characterized by sudden and severe abdominal pain, often in the upper right part of the abdomen. The pain from pancreatitis can be caused by different mechanisms, such as abnormal activation of pancreatic zymogens or NF-κB mediated inflammation in the pancreas. The treatment of pancreatitis depends on its type, severity, and underlying cause. Hospitalization and medications are typically necessary, while in others, surgery may be required. Proper management of pancreatitis is essential, as it can help reduce the risk of complications and improve the patient's quality of life. The literature on pancreatitis pain management evaluates systematic approaches and the effectiveness of various treatments, such as lidocaine, opioid agonists, ketamine, magnesium, endoscopic methods, spinal cord stimulation, and other novel treatments present opportunities for exploration in pancreatitis pain management.

**Keywords:** acute pancreatitis; pain; management; treatment

## 1. Introduction

Pancreatitis is a condition that involves inflammation of the pancreas, an organ located posterior to the stomach. This condition can be acute or chronic, depending on the duration of symptoms. Acute pancreatitis (AP) is characterized by a sudden onset of inflammation and swelling in the pancreas. However, the most utilized classification systems for acute pancreatitis do not provide information on pancreatitis etiology. The Atlanta classification is a commonly used system for classifying acute pancreatitis, which was most recently revised in 2011. Classification is based on the phase and severity of the disease and considers the presence of complications and organ damage or failure. It includes categories such as early or late phase, and mild, moderate, or severe severity [1].

The most common etiologies of pancreatitis are gallstones and alcohol use, the latter of which accounts for 60–80% of all cases [2]. Other known causes include metabolic hypertriglyceridemia and hypercalcemia, drug-induced, autoimmune, endoscopic retrograde cholangiopancreatography, and idiopathic causes. In remaining patients, physical examination, laboratory tests, imaging studies, and extensive history taking yields unexplained etiology [3].

### 1.1. Epidemiology

The annual incidence of acute pancreatitis in the United States is 3 to 45 per 100,000 persons [4]. For chronic pancreatitis, annual incidence is 5 to 12 per 100,000 and prevalence is 50 per 100,000 persons. Acute pancreatitis occurs equally among males and

females and incidence increases with increasing age. Pancreatitis among males is more likely to be from an alcohol etiology, whereas pancreatitis among females is more likely to be related to gallstones, endoscopic retrograde cholangiopancreatography idiopathic, or autoimmune causes [4]. Pancreatitis and pancreatic cancer are most common among those who identify as Black compared to any other race/ethnicity group.

The incidence of acute pancreatitis in the United States has increased over recent years. Acute pancreatitis is the most common gastrointestinal reason for hospital admission in the United States and is associated with a 1% mortality in mild cases and up to 10–20% in severe cases [4,5]. A recent systematic review on acute pancreatitis found a global increase in risk factors for acute pancreatitis such as age, biliary disease, obesity, metabolic syndrome, and alcohol use. Obesity is a risk factor for both increased incidence and increased severity of acute pancreatitis. Increased rates of obesity promote the formation of gallstones and result in elevated levels of triglycerides [6]. Obesity also increases the severity of acute pancreatitis through a pro-inflammatory increase in unsaturated fatty acids which inhibit mitochondrial complexes and result in necrosis [6].

### 1.2. Morphology

The pancreas is a gland located in the abdomen that has both exocrine and endocrine functions and it produces hormones such as insulin, glucagon, somatostatin, and synthesizes digestive enzymes. The pancreas is divided into four regions: head, neck, body, and tail. The head is in the right upper abdominal quadrant, near the duodenum, while the tail extends into the hilum of the spleen in the left upper quadrant. The pancreas exists mainly in the retroperitoneal space, with the tail being intraperitoneal due to its location within the splenorenal ligament [7].

The pancreas is surrounded by a thin layer of connective tissue that forms a seemingly lobulated capsule around the gland and penetrates it as various septa. Each lobule is composed of serous secretory units called acini, which form exocrine portions of the pancreas and empty their secretions through pancreatic ducts. Each acinus is made up of clusters of simple epithelial cells called pyramidal serous cells, which surround a central duct lumen lined with low simple cuboidal epithelium [8]. As ducts increase in size, this low cuboidal epithelium is replaced with stratified cuboidal epithelium [9].

### 1.3. Endocrine and Excocrine Function

The islets of Langerhans, which are responsible for endocrine functions of the pancreas, are dispersed between acini. Islets are composed of three main cell types: alpha cells, which secrete glucagon; beta cells, which secrete insulin; and delta cells, which secrete somatostatin. Islet cells have a polygonal shape and are arranged in short, irregular cords that are intertwined within a network of capillaries. Islets vary in size and are more concentrated in the pancreatic tail. Other hormones such as VIP, pancreatic polypeptide, motilin, serotonin, and substance P are also produced by cells in islets [10–12].

The main pancreatic exocrine secretions are drained through smaller ducts that begin in the tail. These ducts merge into the main pancreatic duct, which runs along the posterior surface of the body and neck. In the head, the main pancreatic duct is located left of the common bile duct from the liver and gallbladder. These two ducts enter the duodenum together at the hepatopancreatic ampulla of Vater, which is surrounded by the sphincter of Oddi. Additional drainage of the pancreatic head is provided by the accessory pancreatic duct, which opens into the duodenum at the minor papilla [13].

### 1.4. Vascularization and Lymphatics

The pancreas is supplied with blood from branches of the celiac trunk and superior mesenteric artery. The majority is supplied by pancreatic branches of the splenic artery, a branch of the celiac trunk. The head of the pancreas also receives blood from the superior and inferior pancreaticoduodenal arteries, which are branches of the gastroduodenal artery and the superior mesenteric artery, respectively. Venous drainage is performed by the

hepatic portal system via the splenic vein and superior mesenteric vein. The head and neck of the pancreas drain into the superior and inferior pancreaticoduodenal veins, which partly drain into the portal vein and partly into the right gastroepiploic vein. Lymphatic drainage follows pancreatic arteries, with the body and tail draining into retropancreatic nodes and the head and neck draining into celiac and superior mesenteric nodes [14,15].

## 2. Pathophysiology of Acute Pancreatitis Pain

Precise causes of acute pancreatitis are not fully understood. However, researchers have identified acinar cells as primary drivers of pancreatic injury and have studied cellular changes that occur during the progression of the disease. Two main pathophysiological pathways have been identified in early pancreatitis, one involving abnormal activation of pancreatic enzymes, and the other an inflammatory cascade mediated by NF-κB (Figure 1). Destruction of the pancreas and systemic inflammation are the main characteristics of pancreatitis. The main causes of destruction include premature activation of trypsinogen, and an enzyme within acinar cells, instead of the duct lumen. This is typically triggered by an increase in ductal pressure and problems with calcium homeostasis and pH. Many toxins that cause pancreatitis lead to ATP depletion, which increases intra-acinar calcium concentrations and stimulates the early activation of trypsinogen to trypsin, activating other enzymes such as elastase and phospholipases.

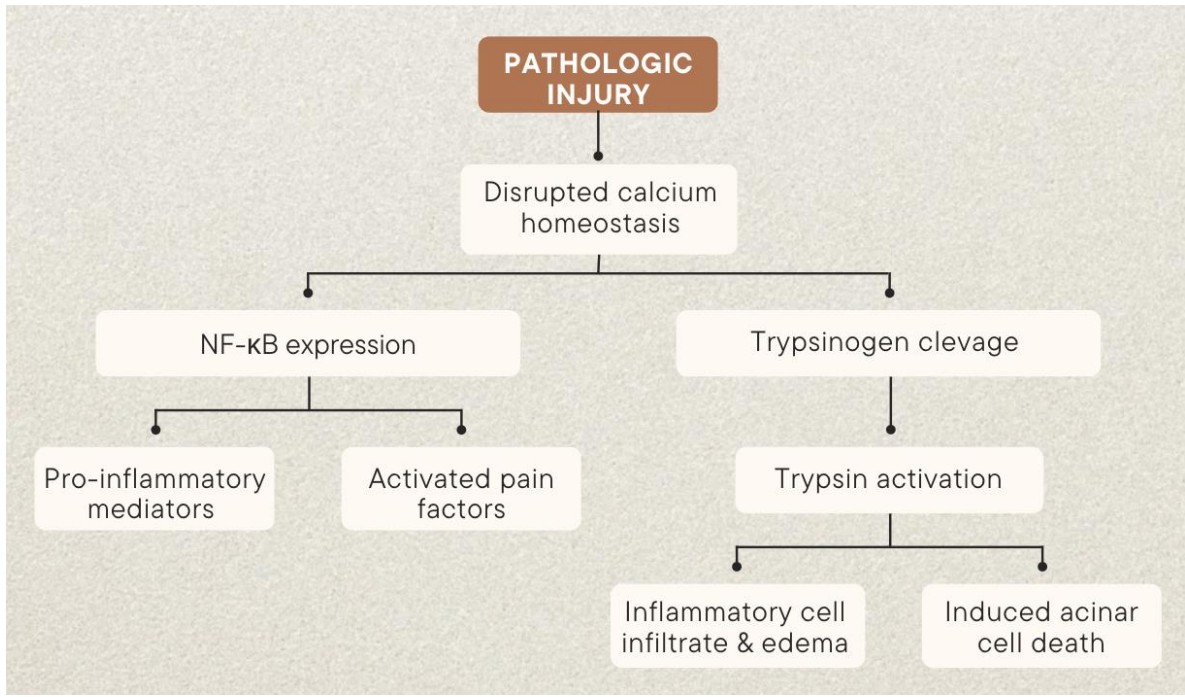

**Figure 1.** Two critical autonomous and parallel pathways in early pancreatitis [16].

### 2.1. Trypsin

Aberrant pancreatic enzyme activity has long been implicated in the pathogenesis of pancreatic injury. Subsequent investigation has implicated trypsin, a product of pancreatic acinar cells typically secreted into the pancreatic duct, in early pancreatic damage. Under physiologic conditions, the proenzyme trypsinogen is maintained in the pancreatic milieu and activated to trypsin only in the duodenum, where it serves as a key digestive protease. Several regulatory mechanisms serve to prevent deleterious effects of premature activation. Chiefly, geographic separation of trypsinogen from its primary activator restricts premature activity. In non-diseased states, trypsinogen is activated by enterokinase, a specialized serine protease restricted to the duodenal brush boarder [16]. Several mechanisms prevent trypsinogen activation. Cysteine protease cathepsin L promotes degradation of intra-acinar

trypsinogen while chymotrypsin C will inhibit aberrantly activated trypsin within the parenchyma [16,17].

Activation of trypsin in early acute pancreatitis has been extensively studied in animal models. Trypsin activity was found to rapidly increase after the administration of caerulein, a common toxin used to replicate pancreatitis in mouse models [18]. Trypsin activity has also been re-demonstrated in other models of pancreatic injury, e.g., in both iatrogenic-induced injury and knockout models of inherited pancreatitis [19,20].

### 2.2. Calcium Signaling

Calcium signaling has been identified as a likely nidus for trypsin activation. Transient spikes in intracellular calcium concentrations play a role in normal acinar function [21,22]. Cholecystokinin (CCK) is an important activator of acinar synthetic function. It acts by binding to G-protein-coupled receptors, which initiate a cascade that leads to calcium efflux from the endoplasmic reticulum to the cytoplasm [18,23]. In pathological states, sustained global increases in intra-acinar cytoplasmic calcium are observed in early acute pancreatitis. This becomes apparent in states of hypercalcemia, which is characterized by high blood levels of calcium. Hypercalcemia can lead to formation of pancreatic calcium deposits which can impair function and increase the risk of inflammation. Further, hyper-calcemia can exacerbate inappropriate activation of digestive enzymes, causing damage and leading to pancreatitis. Indeed, calcium blockade in mouse models has a protective effect against acute pancreatitis. Endoplasmic reticulum surface ryanodine receptors (RyR) and membrane-bound Store operated channels (SOC) are the two most common sources of calcium influx [18,24]. Absence of RyRs and SOCs in separate mouse models have a significant protective effect against acute acinar cell injury [22,25,26].

### 2.3. Calcineurin

Globally elevated levels of cytoplasmic calcium likely have a diverse range of effects. However, calcineurin has been identified as mediator between calcium and premature trypsin activation. Calcineurin is a calmodulin-activated phosphatase with a calcium-dependent regulatory subunit that has catalytic effects in numerous cell types [27–29]. Within acinar cells, calcineurin overexpression has been shown to potentiate acute pan-creatitis independent of calcium levels [27,29]. Calcineurin inhibition in mouse models has a protective effect against zymogen activation and acute pancreatitis without affecting calcium concentration [30]. Thus, calcineurin appears to play a role in the downstream of calcium influx, but also in the upstream of trypsin activation in acute pancreatitis. Crucially, calcineurin has been directly implicated in the propagation of ethanol and bile acid-induced pancreatitis—two leading causes of acute disease [31].

### 2.4. Co-Localization and Autophagy

Abnormal organelle co-localization, including secretory vacuoles and lysosomes, has been well documented in early acute pancreatitis following calcium influx [32–34]. The fusion of vacuoles with lysosomes facilitates the activation of trypsinogen by lysozymes. The lysosomal protease cathepsin B has been shown to play a role in the premature trypsinogen activation and progression of acute pancreatitis [35,36]. Cathepsin B knockout models in mice show reduced rates of acute pancreatitis in response to acinar injury, while inhibition of cathepsin B results in decreased trypsinogen activation [27].

### 2.5. Inflammatory Pathway

Acinar cell injury initiates an inflammatory response, which runs in parallel to trypsino-gen activation. Studies in trypsinogen knockout models show persistent inflammatory damage, suggesting a secondary mediator of pancreatitis. NF-κB is currently the most described inflammatory mediator active in early pancreatitis [37–40]. The NF-κB cascade is initiated by calcium signaling, but it is also related to other, less described signals, including protein kinase c [40,41]. This cascade runs in parallel, but independently from trypsinogen

activation [42,43]. However, the exact mechanisms of NF-κB activation remain unclear, and a variety of other potential mediators have been suggested. Despite this uncertainty, acinar models where NF-κB is overexpressed demonstrate significant cellular inflammation that is sufficient enough to induce systemic inflammation [18,44]. Ultimately, the activation of acinar inflammatory cascades leads to oxidative stress and apoptosis [24].

### 2.6. Neuropathic Pain

Neuropathic pain in pancreatitis relates to several neuronal alterations in the setting of acute inflammation. It is classically described as sharp, intense, and hot pain, and several questionnaires have been designed to identify and direct treatment for patients with prominent neuropathic pain [45–48]. Neuropathic pain is caused by damage to the peripheral or central nervous system. This can result in the damage of nerve endings, leading to the development of neuropathic pain states. In pancreatitis, neuropathic pain is thought to play an important role in the development of pain. Damage to nerve endings and the sensitization of these nerves via inflammation in acute pancreatitis may lead to the development of neuropathic pain [45,49–54].

There are several mechanisms involved in the development of neuropathic pain. These include peripheral and central sensitization. Peripheral sensitization occurs with changes to the uninjured primary afferent nociceptor after nerve lesions. This results in sensitization and ectopic spontaneous activity of primary afferent nociceptors. Some changes that may be involved include the expression of voltage-gated sodium channels, transmembrane proteins such as vanilloid receptors and temperature-sensitive excitatory ion channels, and adrenoreceptors on uninjured primary afferent nerves [55,56].

Central sensitization, on the other hand, occurs with change in the central nervous system. This can result in the development of abnormal pain sensitivity and heightened pain responses to normally non-painful stimuli. Factors that may contribute to central sensitization include nerve growth factors, pro-inflammatory cytokines, and changes in the expression of certain receptors and ion channels. Leukocytic invasion of myelin and Schwann cells results in a pancreatic neuritis contributing to persistent pain [51]. Glial cell proliferation in response to injury has also been correlated with increased pain [50,57,58].

### 2.7. Inflammatory Pain

Inflammatory pain is a type of nociceptive pain which is caused by tissue damage or inflammation [59]. Inflammatory pain involves inflammatory mediators that activate and sensitize nociceptors which then activate neurons to relay signals [59]. Pancreatic sensory innervation is mostly sympathetic; cell bodies of pancreatic afferent nerves are located in the dorsal root ganglion (DRG) at T5-L2 and are activated in AP [60]. When mechanical or chemical stimuli destroy acinar cells, digestive enzymes and increased inflammatory mediators are released in the vicinity [61]. These mediators include K+, H+, ATP, histamine, substance P (SP), bradykinin, prostaglandins, and norepinephrine, all of which further amplify the inflammatory response [61,62].

Along with histamine, the activation of mast cells causes the release of nerve growth factor (NGF), promoting immune cell recruitment, sensitization of nociceptors, and plastic changes in neurons that contribute to pain and inflammation [60,61,63]. Increased levels of mast cells have been found in tissue specimens of patients with pancreatitis [64]. NGF is also upregulated in pancreatitis, and it itself upregulates transient receptor potential cation channel subfamily V, member 1 (TRPV1), SP and calcitonin gene-related peptide (CGRP) [65]. TRPV1 is a nociceptive receptor implicated in pancreatitis that responds to stimuli including inflammatory molecules, heat, and acidosis [65]. In animal models, a lack of TRPA1/TRPV1 activity is associated with decreased fibrosis, inflammation, neural hypertrophy, and sensitization to pain [65].

SP and CGRP are neuropeptides that are released from afferent nerve endings in neurogenic inflammation and are increased in AP; TRPV1 activation results in SP release [60,65–67]. SP works through the neurokinin-1 (NK1) receptor in acinar cells, leading to neurogenic

inflammation and pain via increased pro-inflammatory mediators, nuclear factor kappa B (NF-κB) activation, and necrosis in murine models of AP [62]. NF-κB induces the transcription of more proinflammatory mediators that recruit leukocytes following acinar cell injury [62]. In mice with cerulein-induced AP, the NF-κB pathway in the DRG was activated by nitric oxide (NO), which was elevated in AP alongside the increased iNOS (nitric oxide synthase) expression and promoted expression of pain factors [67]. Conversely, levels of kappa opioid receptor (KOR) decreased [67]. Furthermore, a NO donor increased SP and CGRP in the DRG and decreased the expression of Oprk1 (which encodes KOR), while the opposite effects were observed with a NO scavenger [67]. iNOS/NO may be located upstream of NF-κB, which may be upstream of the KOR [67].

The bradykinin receptor is involved in pain transmission and is mediated by kinins, which are peptide hormones that affect the regulation of blood pressure, pain sensitization, and cell growth [62,68]. Bradykinin and activated proteases are released following acinar cell necrosis in AP, leading to a 'necrotic amplification loop' with neighboring stellate cells through the activation of NO via $Ca^{2+}$ signals [62]. Bradykinin may also activate the endings of afferent neurons, resulting in SP and CGRP release and neurogenic inflammation [60]. The application of NGF, bradykinin, histamine, 5-HT, and CCK octapeptide to a rat pancreas in electrophysiological studies has been shown to induce excitatory and inhibitory afferent responses [60]. Galanin is another neuropeptide involved in AP and is associated with pain through effects such as neuroinflammation, enhanced neutrophil function, and inhibition of ductal $HCO_3^-$ secretion [62]. Additionally, the inflammatory cytokine prostaglandin E2 (PGE2) acts on stellate cells to regulate pancreatic fibrosis, and its involvement in nociceptive signaling may contribute to severe pain in pancreatitis that does not correlate with structural changes [65]. Finally, transforming the growth factor beta (TGFβ) causes nociceptive sensitization and contributes to pain and inflammation in pancreatitis, possibly by upregulating artemin in stellate cells [65].

### 2.8. Visceral Pain

Visceral pain also falls under nociceptive pain and is transmitted by afferent sympathetic fibers and the vagus nerve [61]. Nociceptive fibers from primary afferent nociceptors synapse with second order neurons at the dorsal horn, sending signals through ascending pathways to thalamic, limbic, and cortical brain structures [61,65]. Specifically, pain is transmitted to the thalamic ventral posterolateral (VPL), ventromedial, intralaminar central lateral (CL), and parafascicular nuclei, as well as areas in the brainstem [69]. After pain processing takes place, efferent impulses travel to the pancreatic structures and immune cells to illicit a response [61]. Visceral pain is not well-localized and is felt as aching, burning, and cramping [61]. It may be referred to the body wall at the level of innervation of the nociceptive stimulus [61]. In pancreatitis, these areas of referred pain become larger, and each nerve fiber may correspond to multiple neighboring organs in addition to the pancreas [65]. This overlap of sensory innervation makes it difficult to distinguish pain associated with pancreatitis from other sources of abdominal pain [65]. Visceral hypersensitivity and neural sensitization are now believed to contribute more to pancreatitis pain than ductal obstruction, thus explaining why pain may not improve after surgery [70]. In specimens from humans with pancreatitis, the increased expression of growth-associated protein 43 (GAP-43), NGF, its receptor TrkA, and brain-derived neurotrophic factor correlate well with pain scores [64].

Visceral hypersensitivity in pancreatitis has been demonstrated in rat models, which show an increased activity in dorsal root ganglia and a relatively depolarized resting state, as well as an increased expression of TRPV1 compared to controls [64,71,72]. TRPV1 and TRPV4 antagonists have been shown to decrease visceral pain responses in rat models of pancreatitis [64,73]. Furthermore, decreased glutamatergic synaptic strength in the midbrain ventrolateral periaqueductal gray (vlPAG), indicated by a reduced AMPA/NMDA ratio, has been found in rats with dibutyltin dichloride (DBTC)-induced visceral pancreatic pain [74]. The vlPAG is responsible for antinociception via descending pain inhibition, so de-

creased synaptic transmission in this region results in a decreased inhibition of descending pain and thus abdominal hypersensitivity, which is alleviated by intra-vlPAG microinjection of AMPA [74]. Activated microglia, which release pro-inflammatory cytokines and chemokines resulting in altered neuronal activation and central sensitization, were found in the CL thalamus and other pain-responsive brain regions of mice with caerulein-induced recurrent AP [69]. KOR may also play a role in visceral pain processing, indicated by an increased response to visceral chemical pain found in KOR-deficient mice [67,75]. In a study that was previously discussed under the inflammatory pain section, mice with AP had a decreased KOR expression, suggesting that KOR is involved in AP-associated visceral pain [67]. Trypsin may also contribute to pancreatic hypersensitivity via the activation of protease-activated receptor-2 (PAR-2) on peripheral sensory neurons [64,76]. The infusion of trypsin at sub-inflammatory concentrations and PAR-2-specific activating peptide (AcPep) into rat pancreatic ducts produced similar behavioral pain responses. The demonstration of pharmacological desensitization provided further evidence of a shared mechanism of PAR-2-mediated nociceptive effects by trypsin and AcPep [76]. The same study found that 98% of the pancreas-specific DRG neurons expressing TRPV-1-IR (a marker for nociceptive neurons) also expressed the PAR-2 receptor, suggesting that the most visceral afferent nociceptive nerve fibers express PAR-2 [76].

### 2.9. Gut Flora Dysbiosis

The human gastrointestinal tract is home to a vast community of microorganisms, which play a crucial role in the health of the human host and also contribute to numerous disease processes. It is estimated that over $10^{14}$ microorganisms reside within the human gastrointestinal tract [77]. Some authors have referred to the human microbiome as a hidden "metabolic organ", due to its integral role in host metabolism, physiology, nutrition, and immune function [78]. The role of the gut microbiome has been implicated in a variety of disease processes, including inflammatory bowel disease [79], irritable bowel syndrome [80], colon cancer [81], Alzheimer's Disease [82], asthma [83], obesity [84], and autoimmune disorders [85,86].

Over the past two decades, a myriad of research efforts have gone into investigating the relationships between human microbiomes and the pathogenesis of acute pancreatitis (Table 1). As the intestinal mucosal barrier is impaired in more severe forms of AP, an opportunity arises for bacterial translocation via hematogenous, lymphatic, and reflux [87]. Bacterial translocation results in a SIRS response and can compound inflammatory responses seen in AP and result in secondary infection [88]. The human microbiome has been implicated in underlying mechanisms leading to mucosal injury and bacterial translocation in AP. Major end metabolic products of gut microbiota are short-chain fatty acids (SCFA) [89] with acetate, propionate, and butyrate being the most abundant SCFA produced [90]. SCFA are the main source of energy for intestinal epithelial cells and they play a crucial role in maintaining the integrity of the intestinal mucosal barrier [91]. SCFA also reduce the pH of the intestinal tract, creating an environment more conducive to commensal bacteria growth and inhibiting the growth of pathogenic species, including *Escherichia coli* and *Shigella* [92]. In humans, acetate and propionate are largely produced by *Bacteroidetes* species, while *Firmicutes* species are the main contributors of butyrate [93]. Zhu et al. demonstrated in rat models that decreased *Firmicutes* (and therefore butyrate production) was associated with an increased inflammatory response and progression to necrotizing pancreatitis in AP [94]. Compared with healthy volunteers, patients with AP were more likely to have gut dysbiosis, with higher populations of *Enterobacteriaceae* and *Enterococcus* and lower *Bifidobacteria* [95]. Additionally, gut dysbiosis has been shown to correlate with the severity of AP, as those with more severe forms of AP were shown to have a reduction in commensal bacteria including *Bacteroides*, *Alloprevotella*, and *Blautia* compared to more mild forms of AP [96].

**Table 1.** Pancreatitis flair pathophysiology.

| Pancreatitis Flair Pathophysiology | |
|---|---|
| Trypsin | Rapidly increasing trypsin, a product of pancreatic acinar cells typically secreted into the pancreatic duct, induces early pancreatic damage [18] |
| Calcium Signaling | Endoplasmic reticulum surface ryanodine receptors (RyR) and membrane-bound Store-operated channels (SOC) increase acinar cytoplasmic calcium [22,25,26] |
| Calcineurin | Calcineurin plays a role downstream of calcium influx and upstream of trypsin activation in acute pancreatitis [27–29] |
| Colocalazation and Autophagy | Vacuole fusion with lysosomes activates trypsinogen—protease cathepsin B has been shown to play a role in premature trypsinogen activation [35,36] |
| Inflammatory Pathway | NF-κB cascade is initiated by calcium signaling and leads to increasing oxidative stress and apoptosis [37–40] |
| Neuropathic Pain | Dilation of intraparenchymal nerves with background fibrosis suggests that increased axonal diameter is likely responsible for initiating neuropathy [49–54] |
| Inflaammatory Pain | Stimuli destroy acinar cells and digestive enzymes and inflammatory mediators are released in the vicinity including K+, H+, ATP, histamine, substance P (SP), bradykinin, prostaglandins and norepinephrine [60,61,63] |
| Visceral Pain | Increased activity of dorsal root ganglia and relatively depolarized resting state and increased expression of TRPV1 [64,71,72] |
| Gut Flora Dysbiosis | Patients with AP are more likely to have gut dysbiosis, with higher populations of *Enterobacteriaceae* and *Enterococcus* and lower *Bifidobacteria* [95] |

The use of probiotics in the treatment of AP has been controversial. Previous animal studies have suggested that probiotics may reduce intestinal mucosal barrier injury and prevent bacterial translocation and secondary infections [97,98]. However, the PROPATRIA trial (probiotic prophylaxis in patients with predicted severe acute pancreatitis), published in 2008, demonstrated a significantly higher mortality rate in patients receiving probiotics (16% vs. 6%) [99]. More recent trials have not shown any negative consequences associated with the use of probiotics [100,101]. A 2015 Cochrane review concluded that further studies are needed to assess the safety and efficacy of probiotic use in AP [102].

## 3. Initial Treatment

Acute pancreatitis requires proper care and management. Nutrition, intravenous fluids, and immediate pain medication are acutely essential for proper treatment. Upon admission to the hospital, treatment should be tailored to the severity of the disease. It is recommended to have a low index of suspicion for 3D abdominal imaging to avoid missing complications that may lead to further compromise. If necessary, referral to specialist or tertiary services may be needed for complex pancreatic disease. To prevent recurrence and properly address the aftermath, continued care is necessary.

### 3.1. Nutrition

Acute pancreatitis causes a hypermetabolic state, which is characterized by an increase in lipolysis, protein breakdown, insulin resistance, and weight loss. These effects are more extreme in severe cases of the disease, and they can be exacerbated by poor nutrition and infection [103]. Early oral refeeding is typically recommended as soon as it is tolerated. If this is not possible, liquid food supplements or enteral tube feeding should be given within a day or two of admission [104].

This may not be necessary in mild cases of acute pancreatitis. The nasogastric route is easier to use than the nasojejunal route, but some patients may require the latter if they are unable to tolerate the former due to delayed gastric emptying. Compared to parenteral nutrition, oral or enteral feeding is associated with lower pro-inflammatory responses and a reduced likelihood of bacterial translocation across the gastrointestinal permeability barrier.

However, enteral tube feeding is limited in patients who are hemodynamically unstable, have gastrointestinal intolerance, or require frequent interruptions for investigations or interventions [105]. This can lead to delays in providing nutritional support.

Attempts to maximize enteral nutrition should be avoided in patients who are not volume-replete, as this can increase the risk of gut injury through non-occlusive mesenteric ischemia. Inadequate nutrition has led to the use of combined enteral and parenteral nutrition, with the latter being started before, during, or after enteral intake is considered insufficient [106,107].

*3.2. Intravenous Fluids*

In acute pancreatitis, the immediate administration of intravenous fluids is crucial for correcting volume loss and tissue hypoperfusion, which can help counteract pancreatic and systemic microcirculatory impairment caused by various inflammatory processes [108]. Early intravenous fluid resuscitation within 24 h of disease onset can lead to lower rates of persistent systemic inflammatory response syndrome and organ failure. It is recommended that fluids are administered at a rate of 5–10 mL/kg/h [109]. The goals of fluid resuscitation are to decrease and/or maintain the heart rate at less than 120 beats per minute, measure urine output through a catheter at more than 0.5 mL/kg/h, and if non-invasive continuous arterial pressure measurement is available, maintain a mean arterial pressure of 65–85 mm Hg with a hematocrit of 35–44% [109].

In critically ill patients, invasive monitoring may include determination of stroke volume variation and intrathoracic blood volume. In mild and moderately severe cases, organ dysfunction is likely to resolve with intravenous fluid resuscitation. However, in severe disease, there is a significant risk of excessive intravenous fluid therapy. Fluid overload in the critical care setting is associated with an increased risk of death due to the negative effects on all major organ systems [110]. It is important to assess the patient's responsiveness to continually higher fluid rates, when necessary, but not when there is a risk of overload. Caution is necessary as clinical assessment has its limitations, and passive leg raising followed by the measurement of cardiac parameters may be an effective method for assessment, but this requires further confirmation [111].

*3.3. Pain Management*

There are many options for managing the pain caused by acute pancreatitis, but the current opinion within gastroenterology follows four principles outlined by Pandanaboyana et al. [112]:

1.    Tailor treatment to the individual as one size will not fit all;
2.    Assess pain intensity and pancreatitis severity to select analgesics;
3.    Step-down approach required for prompt pain relief;
4.    Use opioid-sparing strategies wherever possible.

Abdominal pain is a common symptom of acute pancreatitis and is usually treated with strong opioid medication to reduce the need for other types of pain relief. This can be supported by using pain management strategies, such as pain ladders, for those with less severe pain. Non-steroidal anti-inflammatory drugs (NSAIDs) may be used as an alternative for mild cases, but they can cause kidney damage in more severe disease [113].

Some evidence suggests that opioids can increase pressure in the bile duct, but studies have shown that they are as effective and safe as other options in treating acute pancreatitis [114,115]. Providing early oral nutrition that does not worsen abdominal pain may help reduce pain intensity and duration, as well as the need for pain medication and the risk of food intolerance. In more severe cases, recurring or worsening pain when eating may delay the return to a normal diet. The intensity and duration of pain are generally proportional to the severity of the disease and the total amount of opioid medication used. Continuous intravenous opioid infusions may be used for persistent, severe pain, but the benefits of patient-controlled versus nurse-controlled administration are unclear [116].

Epidural anesthesia has not been found to be significantly better than other options in small trials, but larger studies may determine if it provides additional benefits, such as improved pancreatic blood flow [117]. Pain management should be part of a comprehensive approach that addresses physical and psychological issues and involves patients in their care. Opioids should not be used as a substitute for providing patients with information about their illness, managing their expectations, and offering psychological support to reduce anxiety, especially for younger patients [113,118].

## 4. Pharmacological Treatment

The pain associated with acute pancreatitis is caused by multiple mechanisms indicating a range of analgesics that may be effective in management. When selecting a treatment, it is important to consider the potential for the development of chronic pain (Table 2).

### 4.1. Lidocaine

Systemic lidocaine has been shown to reduce pain in perioperative settings through multiple meta-analyses [119–121], though the degree of efficacy still remains somewhat in question due to variability and quality of previous studies [122]. The exact mechanism of action behind lidocaine and other local anesthetic antinociceptive properties remain partially unclear given their interactions with a large variety of ion channels as well as intracellular second-messenger pathways. Traditionally, lidocaine's antinociceptive properties are believed to have resulted from a blockade of voltage-gated Na+ channels (VGSC's). VGSC's have been repeatedly shown to play a crucial role in the development of neuropathic and inflammatory pain signals; however, lidocaine's effects on potassium channels, calcium channels, Gaq-coupled receptors, NMDA receptors the glycinergic system, and serotonin receptors cannot be ignored, and pain modulatory effects are likely multimodal. Furthermore, lidocaine has also been shown to have anti-inflammatory properties as well, decreasing leukocyte activation, adhesion, and migration, though these mechanisms are incompletely understood [123]. Interestingly, multiple studies have also demonstrated a dose-dependent relationship between lidocaine infusion and vascular effects, with lower concentrations producing vasoconstriction and higher concentrations producing vasodilation [124].

Currently, no studies have evaluated the efficacy of lidocaine infusion in relation to pain associated with pancreatitis, though some studies have evaluated lidocaine's use for pancreatitis management. Pancreatitis is a common adverse effect of ERCP, with some studies showing 5–10% of patients experiencing this result following the procedure [125]. Lidocaine has been shown to block intramural neural reflexes in the Sphincter of Oddi [126]. In a randomized control trial of 326 patients, topical lidocaine administered to duodenal papilla prior to cannulation vs. a control group found no significant reduction in ERCP-related acute pancreatitis [127]. Another study looking at cerulein-induced acute pancreatitis in rats sought to identify the efficacy of intra-arterial lidocaine infusion [128]. The pancreas is highly sensitive to ischemia, and pancreatitis leads to a decrease in perfusion as a result of multifactorial vessel injury, with increased vascular permeability, edema, and vasoconstriction all playing a role in decreasing blood flow to the pancreatic tissues [129]. The study attempted to assess if lidocaine-induced vasodilation, along with anti-inflammatory properties, could potentially have a beneficial effect in pancreatitis management. Rats who received intra-arterial lidocaine demonstrated statistically significant decreases in amylase and lipase serum concentrations, indicating a potential reduction in pancreatitis severity in treatment groups [128]. While this study potentially indicates a new avenue for pancreatitis management, human trials are required to determine efficacy.

**Table 2.** Pancreatitis pain treatment [128–173].

| Trials Evaluating Pancreatitis Pain Pharmalogical Treatment | |
|---|---|
| Lidocaine | Animal studies indicate intra-arterial significantly decreases amylase and lipase serum concentrations. RCT show topical administration was comparative to controls [128] |
| Hydromorphone/Opioid Agonists | RCT's demonstrate no difference in pancreatitis complications or adverse events between opioid and non-opioid analgesics [114,145,146] |
| Ketamine | Case studies demonstrate success in the treatment of pancreatic pain in both quick and long-lasting analgesia. RCT's are ongoing [155] |
| Magnesium | There is currently one clinical trial investigating magnesium sulfate's effects on preventing post-ERCP pancreatitis [173] |

While few studies have looked at lidocaine for the management of pain in pancreatitis, other studies have sought to identify the utility of other local anesthetics in acute pancreatitis. Jakobs et al. looked at procaine infusion compared to buprenorphine in the management of patients with acute pancreatitis, with the buprenorphine group reporting decreased pain levels and requiring less additional analgesics [130]. A 2004 randomized control trial of 107 patients also evaluated procaine efficacy in acute pancreatitis, and it demonstrated larger pain scores and larger requirements of additional analgesics in patients treated with procaine vs. pentazocine [131]. Conversely, Layer et al. did show some utility of procaine in pancreatitis management when compared with a placebo, with statistically significant reductions in VAS scores (and the use of additional analgesics) showing a greater proportion of patients achieving adequate analgesia [132]. However, a meta-analysis from Thavanesan et al. looking at multiple modes of analgesic management in acute pancreatitis showed local anesthetics to have the least efficacy in the reduction of pain scores when compared to other treatment modalities [115]. More information is needed on whether systemic local anesthetic usage could potentially have some purpose as part of a multi-drug approach to pain management, but current evidence seems to suggest little benefit as a primary pain-controlling agent.

Local anesthetics have also been explored for nerve blocks in pancreatitis management. Epidural anesthesia has shown a significant effect in the reduction of pain scores and the reduction of mortality in acute pancreatitis. This effect is multifactorial, and likely related to the anti-inflammatory effects of locally acting anesthetics along with sympathetic nerve blockade, which is achieved with epidural anesthesia. This sympathetic blockade allows for the redistribution of splanchnic blood flow and thus increases perfusion of pancreatic areas which have been subject to ischemia [118,133]. The transversus abdominus plane (TAP) block has also been used in acute pancreatitis pain management. Landy et al. reported a case of opioid-resistant acute pancreatitis with successful pain management achieved through bilateral TAP blocks [134]. Smith et al. also reported numerous cases where the TAP block served benefit in the management of pain associated with pancreatitis [135]. Interestingly, the TAP block itself is typically associated with a higher degree of somatic analgesia, and significant visceral coverage would be unexpected [136]. More recently, Abdelghafour et al. utilized an erector spinae plane (ESP) block to achieve adequate analgesia in a patient with pancreatitis [137]. Given the somatic, as well as visceral and sympathetic coverage demonstrated in ESP blocks, blocks targeted in lower thoracic vertebrae could be expected to have potentially beneficial effects in the pain management of acute pancreatitis. Given the relative ease of performance and safety of ESP blocks compared to epidural blocks, ESPs could potentially be of significant use for pancreatitis-related pain management in the future, particularly in opioid-tolerant patients [138].

*4.2. Hydromorphone/Opioid Agonists*

Patients with acute pancreatitis are often presented with severe upper abdominal pain that may radiate to the back [139]. As such, pain management is a critical component in treating acute pancreatitis. Opioids have been shown to be effective and well-tolerated in

treating pancreatitis-related pain [114]. Opioid receptors are expressed in central and peripheral neurons and modulate pain primarily through three G protein-coupled receptors: mu, delta, and kappa [140]. Mu receptors influence reward, euphoria, sedation, and respiratory drive [141]. Common opioids in clinical practice (morphine, codeine, methadone, fentanyl) are mu receptor agonists. Hydromorphone, a semi-synthetic opioid agonist and hydrogenated ketone of morphine, is also used in treating pancreatitis pain [142]. Hydromorphone has been shown to exhibit a stronger mu receptor binding affinity compared to hydrocodone [143]. Buprenorphine, a partial agonist and partial antagonist at the mu-opioid receptor with a comparatively long duration of effect, has been shown to have analgesic properties, though studies demonstrating its use in pancreatitis are relatively scarce up until this point [144]. In practice, opioids for pancreatitis-associated pain are administered orally or intravenously as either a bolus or infusion.

Interestingly, studies have found opioid and non-opioid analgesics have similar effectiveness regarding pain control, adverse effects, and risk of pancreatitis complications [114,145,146]. As mentioned in a previous section, a systematic review of five randomized controlled trials including 227 patients found that opioids may decrease the need for supplementary analgesia, but there was no difference in pancreatitis complications or adverse events between the opioid and non-opioid analgesics groups [114]. The opioid analgesics analyzed in the systematic review were intravenous and intramuscular buprenorphine, intramuscular pethidine, intravenous pentazocine, transdermal fentanyl, and subcutaneous morphine, and the non-opioid analgesics analyzed were procaine and metamizole [114]. Two additional randomized controlled studies (RTCs) compared pain relief and adverse effects following tramadol versus non-steroidal anti-inflammatory/acetaminophen administration for acute pancreatitis [145,146]. In their study on 90 patients with non-traumatic pancreatitis, Gülen et al. found that mean pain scores between the tramadol, paracetamol, and dexketoprofen groups were similar 30 min after initial administration [145]. A separate RCT on 46 patients compared tramadol and diclofenac and found no significant difference in pain one hour after administration, the number of patients requiring supplemental analgesia, and the number of painful days between the two groups [146]. Pancreatitis-related complications (acute lung injury, pleural effusion, ascites) were lower in the diclofenac group; however, the result was not statistically significant. A prospective randomized trial of 40 patients examining buprenorphine vs. procaine for pain management in pancreatitis showed that buprenorphine resulted in a significant reduction of both VAS scores and the need for rescue analgesia with minimal variation in side effects [130]. Blamey et al. examined pain-free periods in 32 patients being treated with either buprenorphine or pethidine, with no statistically significant difference being found [147].

### 4.3. Ketamine

Ketamine is a non-competitive N-methyl-D-aspartate (NMDA) receptor antagonist. NMDA receptors are excitatory inotropic glutamate receptors found in both the central and peripheral nervous system, and are involved in pain signal amplification, development of central sensitization, and opioid tolerance [148]. Ketamine has been shown to exert its anti-hyperalgesic effect by returning the NMDA receptor to its resting state condition and reducing or reversing opioid tolerance [149,150]. Ketamine exists in two stereoisomeric forms, with the S-isomeric form having four times more affinity for NMDA receptors than the R-isomeric form [151].

NMDA receptor antagonism is responsible for the analgesic effects of ketamine, but interactions with opioid receptors are thought to also play a partial role. Ketamine has been shown to utilize mu, delta, sigma, and kappa opioid receptors [151–153] and interact with nicotinic and muscarinic acetylcholine receptors, monoaminergic receptors, and voltage-sensitive sodium channels [154]. More recently, studies have suggested that ketamine may play a role in modulating pain transmission through limiting activation of astrocytes and microglia [154].

Ketamine use for pain management in pancreatitis has been explored in several studies. A single center study, by Sheehy et al., investigating the use of subanesthetic ketamine for pain management found that ketamine was effective in reducing pain scores in patients with inflammatory diseases, such as pancreatitis [155]. Subanesthetic doses are often adjuvants to opioid therapy in patients with acute and chronic pain demonstrating improved pain scores and decreased opioid intake [156]. Additional data suggest that subanesthetic ketamine for pain management is both feasible and safe in regular care units, and that it benefits children, adolescents, and young adults with both acute and chronic pain [155]. Furthermore, it was found that ketamine infusions in patients with pancreatitis are more often associated with over a 20% reduction in pain scores compared to other types of pain [155]. Bouwense et al. found in a blinded crossover trial that, in patients with pancreatitis, S-ketamine infusions were associated with an increase in pain pressure thresholds [157]. However, the effects were short-lasting and there was no significant effect on VAS scores [157]. A randomized, double-blinded, placebo-controlled clinical trial—titled the RESET Trial—investigating the analgesic and anti-hyperalgesia effects of S-ketamine is currently ongoing [149].

Presently, there are few case reports detailing ketamine use in pancreatitis pain management. Mannion and O'Brien report a case of a patient with severe pain secondary to alcoholic pancreatitis where the subcutaneous infusion of ketamine was found to be an effective and well-tolerated means of pain control [158]. In a case report of a teenager with severe pain secondary to necrotizing pancreatitis, Mulder et al. found that the initiation of ketamine as adjunctive pain management led to a rapid reduction in pain scores by day five of infusion [159]. In another case report of a patient with severe post-ERCP pain secondary to acute pancreatitis, Agerwala et al. found that the initiation of ketamine infusion was effective in producing quick and long-lasting analgesia [160].

The American Society of Regional Anesthesia and Pain Medicine (ASRA) values the use of ketamine because it reduces opioid tolerance and hyperalgesia, reduces opioid requirements, and has the capacity to help manage nociceptive and neuropathic pain. ASRA formulated guidelines for ketamine use, concluding that limited evidence of the benefits of IV made ketamine the essential analgesic for acute or periprocedural pain. However, moderate evidence supports the benefit of ketamine as an addition to opioid based IV treatment for acute and postoperative pain management [156,161]. One advantage ketamine has over opioids for chronic pain is preventing/reversing opioid hyperalgesia and tolerance [162]. The phenomenon of opioid-induced hyperalgesia caused by opioid exposure leads to nociceptive sensitization and a paradoxical response where patients on opioids may become more sensitive to painful stimuli [163]. Another advantage of ketamine over opioid administration is decreased tolerance and tachyphylaxis risk [164,165] and unique pain control [166].

However, ketamine is not without adverse effects, the most common of which are nausea, vomiting, dizziness, drowsiness, diplopia, dysphoria, and confusion [150]. Ketamine use can cause cardiopulmonary effects, such as arrythmias, bradycardia, hypotension, apnea, and respiratory depression [150].

### 4.4. Magnesium

Magnesium is the fourth most abundant cation in the body and it has been linked to over 300 enzymatic reactions modulating energy metabolism and protein and nucleic acid synthesis [167,168]. The current literature suggests that magnesium levels can be decreased in patients with acute pancreatitis; however, exact mechanisms have not been fully elucidated [169,170]. Magnesium has been shown to be a vasodilator, a cytoprotective agent, anticoagulant, and antioxidant [169]. Data on the effects of magnesium supplementation in patients with acute pancreatitis are currently limited. Studies on animal models have suggested that, as a calcium antagonist in the pancreas, magnesium may be able to counteract the calcium-induced protease activation and necrosis in acute pancreatitis [171,172]. There

is currently one clinical trial in progress investigating the effect of magnesium sulfate in preventing post-ERCP pancreatitis [173].

## 5. Opioid Use in Patients with Pancreatitis

With an increased focus on pain management in the early 2000s, opioid prescribing, use, and misuse experienced a sharp uptick [174]. Though the integration of pain as the fifth vital sign solidified the importance of pain assessment, reliance on opioid usage led to an emergence of the pervasive opioid epidemic. In two decades, there was a 200% increase in opioid-related deaths [174]. Physicians have since been keenly aware of the need to monitor and decrease opioid prescriptions when appropriate. Providers used state prescription drug monitoring programs (PDMPs) more than 910 million times in 2020, markedly increased from previous years [175]. In the last decade, there has been a 44.4% decrease in opioid prescriptions nationwide, with a 6.9% decrease from 2019–2020 (AMA). Despite a downward trend, more than 191 million prescriptions were given to patients in 2017 alone, with increasing rates for subsequent years [176].

Non-opioid therapy is generally preferred for first-line treatment in acute and chronic pain. Per the 2016 CDC guidelines, opioids should be prescribed at the lowest effective dosage, with an assessment of the risks and benefits if increasing the dosage to over 50 morphine mg equivalents or more per day is required [177]. Multimodal and multidisciplinary therapies are regarded as the most effective choice to reduce pain and improve function rather than use of single modalities [177]. Opioids are often used to alleviate acute pancreatitis pain, despite a lack of analgesic guidance from the American College of Gastroenterology, American Gastroenterological Association, and the American Pancreatic Association. A review of five randomized control trials found a decreased need for supplementary analgesia in patients receiving opioids, without a difference in the risk of pancreatitis complications or serious clinically adverse events between opioids and other analgesia options [114]. Though studies have not found a difference between the endpoints for opioid and non-opioid analgesics, there remains a clinical preference for opioids over other modalities. In a cohort study of 4307 patients hospitalized for acute pancreatitis, 79.9% received initial treatment with opioids and 9.6% of those continued to receive opioids 90 to 180 days after discharge [178]. Identification of the population that goes on to become prolonged opioid users after hospitalization is important for preventative and therapeutic patient care, given the potential long-term repercussions.

Analgesic effects of opioids come at a cost, and risks should be taken into consideration. Opioids have the potential to cause tolerance, addiction, hyperalgesia, and withdrawal, all of which are heavily intertwined and further precipitate each other's actions. Effects are further amplified when opioids are used chronically. In patients treated with chronic opioid therapy for chronic non-cancer pain, aberrant drug-related behaviors are reported in 20% of patients, with rates of development of opioid use disorder from 0–50% [179].

Opioid addiction is a chronic mental illness characterized by relapses and remissions, and complicated by symptoms of tolerance and withdrawal. With a prolonged administration of opioids, there is decrease in drug potency and efficacy, as tolerance develops and higher doses are needed. Tolerance is defined by The U.S. Food and Drug Administration as a dose of 60 morphine milligram equivalents daily and is complicated by an increased difficulty in the management of pain and an increase in mortality when patients abstain from use and relapse [180]. Patients who develop tolerance and are hospitalized for acute episodes of pain are more likely to have higher risk of mortality and comorbidities, longer hospital stays, and readmit within 30 days [181]. Paradoxically, overuse or misuse of opioids for pain modulation can cause opioid-induced hyperalgesia (OIH), defined as a decreased pain threshold often resulting from dose escalation. Cohort studies in patients with chronic pain regimens found direct correlations in both opioid dose and the duration of treatment with pain intensity and unpleasantness scores [182].

## 6. Novel Treatments

Due to the challenges of managing pain in pancreatitis, novel strategies continue to emerge as treatment options. As the pathophysiology of pain in pancreatitis is multifactorial, multiple methods have been used to address these factors.

### 6.1. Endoscopic Therapy

Ductal obstruction and hypertension may represent one aspect of pain in pancreatitis [70]. Endoscopic interventions aimed at relieving ductal hypertension, including through stone removal, ductal stenting and dilatation, and sphincter manipulation have all been investigated. Many of these interventions are achieved via endoscopic retrograde cholangial-pancreatography (ERCP), and can also be combined with extracorporeal shock wave lithotripsy (ESWL) for further management of pancreatic obstructions.

Advances in endoscopic technology have allowed for improved access into the pancreatic duct. Specifically, sphincterotomy can be used to introduce instruments into the duct [183]. Afterwards, stent placement can be pursued to further alleviate ductal hypertension [183]. In their long-term study on patients undergoing endoscopic sphincterotomy and stenting, Delhaye et al. displayed clinical success, defined as reduced hospitalizations for pain, in 66% of patients, with 30% of patients not requiring any further hospitalizations [184]. Dumonceau et al. demonstrated endoscopic decompression to reduce pain in 73% of patients, with 54% of patients sustaining pain relief at 2-year follow-up [184]. Success rates for the endoscopic treatment of pancreatitis have been reported between 85 and 95% [185,186]. A more recent application in endoscopy has been the usage of endoscopic ultrasound (EUS). In select patients with a challenging anatomy or with trouble accessing various ducts or papilla, EUS-guided drainage of the pancreatic duct can be considered, as it has been proven to show a high technical success rate and reduction of pain in smaller studies [187]. Cahen et al. evaluated endoscopic vs. surgical drainage of the pancreatic duct, specifically focusing on the sustained pain relief after these procedures, finding that surgical management offered more pain relief. As well as this, it required fewer follow-up procedures [188]. This was further validated by Dite et al., who explored surgical vs. various endoscopic procedures including sphincterotomy, stenting, and stone removal. Their results supported surgical management by providing better pain outcomes compared to endoscopic therapies [189]. However, endoscopic therapy, in contrast to surgery, can be offered to patients with more comorbidities and for those who would typically not be surgical candidates [190].

### 6.2. Pancreatic Enzyme Replacement Therapy

Pancreatic exocrine insufficiency is commonly seen in more than half of patients with acute pancreatitis hospitalizations, but this frequency decreases during follow-up. However, it does persist in a minority of patients, including over half of those with pancreatic necrosis [191]. The pancreas typically secretes between 1 and 2 million units of lipase per day, along with various proteases, carbohydrate hydrolases, lipid hydrolases, and nucleases [192]. However, acute pancreatitis disrupts pancreatic secretion [191].

Pancreatic enzyme replacement therapy is likely to be helpful for patients with moderately severe and severe acute pancreatitis who are receiving oral or enteral nutrition until they have repeated normal levels of fecal elastase-1. This therapy is recommended for such patients on a routine basis, and on a long-term basis for those with over half pancreatic necrosis. The presence of exocrine pancreatic insufficiency, indicated by steatorrhea, warrants the use of pancreatic enzyme replacement therapy after acute pancreatitis of any severity and should continue indefinitely if fecal elastase-1 remains below 100 µg/g [191].

This therapy does not need to be stopped for fecal elastase-1 testing. A standard dose for adult patients using one of the licensed preparations (such as Creon, Nutrizyme, Pancrease HL, or Pancrex V in the UK) is 50,000 units of lipase per meal, with half that amount for a snack. This can be increased if steatorrhea has not sufficiently improved. The total daily dose should be adjusted based on the patient's oral or enteral intake, and in

children, it should be based on a combination of body weight and intake. This is so that a maximum daily dose of 10,000 IU/kg is not exceeded and to ensure satisfactory growth and normal levels of fat-soluble vitamins [191].

## 7. Conclusions

The incidence of acute pancreatitis, a condition associated with severe morbidity, healthcare burden, and costs, has risen notably in recent years. Given that biochemical and pathophysiological mechanisms underlying acute pancreatitis are often multifactorial and remain poorly understood, the management of associated pain remains difficult. Pharmacologically, there are a variety of systemic analgesics which were routinely used in this space, although each offers a range of adverse effects, ranging from end-organ symptoms to opioid dependence. To address these challenges, recent approaches to pain management in pancreatitis have focused on targeting the pancreas itself using pharmacologic or procedural methods. This review provides a comprehensive overview of the pathophysiology of acute pancreatitis, as well as a discussion of both traditional and novel treatment options for managing pain in pancreatitis patients. The benefits, drawbacks, and future directions for research in this area are also explored to inform the development of improved pain management strategies for pancreatitis patients.

**Author Contributions:** J.B., A.K., B.Y., H.R., M.W., M.C., N.S.P., S.C. (Salil Chowdhury), R.E., P.F., S.C. (Sarah Chen) and A.G.: conceptualization, methodology, software, validation, formal analysis, investigation, resources, data curation, writing—original draft preparation, writing—review and editing, visualization, supervision, project administration, funding acquisition. All authors have read and agreed to the published version of the manuscript.

**Funding:** This research received no external funding.

**Institutional Review Board Statement:** Not applicable.

**Informed Consent Statement:** Not applicable.

**Data Availability Statement:** Not applicable.

**Conflicts of Interest:** The authors declare no conflict of interest.

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
