# Peer review of "Comprehensive Review of Acute Pancreatitis Pain Syndrome"

_gastrointestdisord, doi:10.3390/gidisord5020014_

Round 1

Reviewer 1 Report

This manuscript describes comprehensive review of acute pancreatitis, especially in terms of pathophysiology and treatment. The authors cited many references and the contents are very fruitful. A reviewer recommend summarized figures for a pathophysiology of acute pancreatitis / its related pain and each therapy. The figures would be useful for many readers.

Author Response

Q: This manuscript describes comprehensive review of acute pancreatitis, especially in terms of pathophysiology and treatment. The authors cited many references, and the contents are very fruitful. A reviewer recommends summarized figures for a pathophysiology of acute pancreatitis / its related pain and each therapy. The figures would be useful for many readers.

A: We appreciate this issue of an additional figure raised by the reviewer. We have taken the opportunity to incorporate a revised summarized figure featuring the pathophysiology of acute pancreatitis in Table 1 with updated references. We have also added figure 1 to more adequately demonstrate the dual pathway described in the manuscript.

Reviewer 2 Report

1. This review seems to focus on the pain treatment and management of pancreatitis, but it details the anatomy of the pancreas, the pathophysiology of pancreatitis, etc., which is too rich in content, but not prominent enough.

2. There are some spelling mistakes.

3. The distinction between acute and chronic pancreatitis is not only based on time. I think this statement is not precise enough.

4. Intracellular calcium signaling is involved in the occurrence of pancreatitis, and hypercalcemia is a risk factor for pancreatitis. Is there any connection between the two?

Author Response

  1. This review seems to focus on the pain treatment and management of pancreatitis, but it details the anatomy of the pancreas, the pathophysiology of pancreatitis, etc., which is too rich in content, but not prominent enough.

We thank the reviewer for their revisions and agree that the anatomy section is not entirely clear and concise and may benefit from a more narrowed focus. The anatomy section was transformed subsequent smaller sections to break up the richly compacted information. We believe this organizes the section and helps to describe the information more clearly and succinctly. The authors believe that this information is informative and important to begin the discussion of the pathophysiology of acute pancreatitis.

  1. There are some spelling mistakes.

The authors thank the reviewer and agree there are misspellings in sections which have subsequently been fixed. As per reviewer #3 revisions, the authors also decided to revise minor points referenced to reduce redundancy within the paper and enhance clarity.

  1. The distinction between acute and chronic pancreatitis is not only based on time. I think this statement is not precise enough.

The authors agree with the reviewer comments. We decided to remove the focus of chronic pancreatitis within the manuscript to discuss acute pancreatitis more clearly. We believe removing the discussion of chronic pancreatitis further clarifies the focus of the paper to discuss associated acute pancreatic pain.

  1. Intracellular calcium signaling is involved in the occurrence of pancreatitis, and hypercalcemia is a risk factor for pancreatitis. Is there any connection between the two?

We thank the reviewer for their comment. We agree that calcium signaling and hypercalcemia as a risk factor is correlated and was cleverly pointed out. Our revisions incorporate information reflecting this important concept within section 2.2.

Reviewer 3 Report

The presented manuscript is well organized and written. The content is valuable, although some more graphical representation particularly in the context of molecular mechanisms of pancreatitis is indispensable. Manuscript contains a lot of details and summarizing some parts (particularly from chapter 1 and 2) in the form of figures would definitely render the text much easier to follow. 
Regarding some minor points, there are many redundant "the" articles, e.g., "The pancreas", "The precise cause", etc. please correct. Please avoid expressions such as "the best described"- in the context of NFkB,  would suggest  using "the most commonly described". Also, please make sure to make sentences straightforward, e.g., instead of saying "has been a controversial topic" just use "has been controversial" or "during their hospital stay", please use "hospitalization", etc. Also, please add appropriate numbers from the reference list to both tables: 1 and 2.

Author Response

Q:

The presented manuscript is well organized and written. The content is valuable, although some more graphical representation particularly in the context of molecular mechanisms of pancreatitis is indispensable. Manuscript contains a lot of details and summarizing some parts (particularly from chapter 1 and 2) in the form of figures would definitely render the text much easier to follow. Regarding some minor points, there are many redundant "the" articles, e.g., "The pancreas", "The precise cause", etc. please correct. Please avoid expressions such as "the best described"- in the context of NFkB, would suggest using "the most commonly described". Also, please make sure to make sentences straightforward, e.g., instead of saying "has been a controversial topic" just use "has been controversial" or "during their hospital stay", please use "hospitalization", etc. Also, please add appropriate numbers from the reference list to both tables: 1 and 2.

A:

We thank the reviewer for their comment and agree that graphical representation will enhance the manuscript as per reviewer #1 comments as well. The authors also thank the reviewer for comments regarding redundant phrases found in the writing and revised accordingly to make the writing more straightforward. The authors revised the paper to also avoid referenced expressions cluttering the writing. Numbers were added from the reference list to support the information found in tables 1 & 2.

Reviewer 4 Report

Acute pancreatitis is a clinically heterogeneous but potentially fatal condition. Despite the investigation carried out, many questions still remain to be clarified. The paper reviews the pathophysiology of acute pancreatitis and particularly the associated pain. The pain therapeutic modalities currently in use are also reviewed.

The topic is relevant and current, however the writing is, in my opinion, hampered by the inclusion of aspects related to chronic pancreatitis and also on therapeutic aspects not related to pain, creating some confusion and extending the text. The main contribution is the review of pathophysiology, since the issue of pain therapy does not bring much new.

Attached are some additional comments

Author Response

Acute pancreatitis is a clinically heterogeneous but potentially fatal condition. Despite the investigation carried out, many questions still remain to be clarified. The paper reviews the pathophysiology of acute pancreatitis and particularly the associated pain. The pain therapeutic modalities currently in use are also reviewed.

The topic is relevant and current, however the writing is, in my opinion, hampered by the inclusion of aspects related to chronic pancreatitis and also on therapeutic aspects not related to pain, creating some confusion and extending the text. The main contribution is the review of pathophysiology, since the issue of pain therapy does not bring much new. Attached are some additional comments. (We attached this pdf file from Reviewer 4 in email, please check)

We extensively thank the reviewer for their revision and certainly agree that the manuscript benefits from the removal of chronic pancreatitis discussions. We have altered the manuscript regarding the authors highlighted comments within the revision to reduce confusion and enhance clarity of the manuscript.

Reviewer #4 Specific Revisions:

The authors were very conflicted on the suggestion to remove section 3.1 & section 3.2. We do agree that these concepts may not need included, but ultimately decided that these treatments help further demonstrate the complex pathophysiology of the pain syndrome and the reasoning behind treatment modalities. The authors did extensively agree that section 3.4 was beyond the scope and confusing to the manuscript and the section was subsequently removed.

We additionally agree that the referenced paragraph in 6.1 and section 6.3 discussed chronic pancreatitis treatments not needed and removed the material. Regarding section 6.2, we decided to continue to include the referenced material. While the authors agree that Enzyme Replacement (ER) is a form of therapy useful in chronic pancreatitis, we believe there is usefulness in acute pancreatitis as well. Below we reference a study finding positive effects in acute pancreatitis that finds “positive tendency in favor of enzyme supplementation found for quality-of-life parameters in all sub scores,” as well as an ongoing randomized clinical trial evaluating the total benefit of ER. We believe that ER may be an important inclusion in any discussion of acute pancreatitis treatment regimens in the future.

pubmed.ncbi.nlm.nih.gov/24618443 & https://clinicaltrials.gov/ct2/show/NCT05480241

The authors revised the paper according to the highlighted comments found within the attached version by reviewer #4. The additions are found in red highlight while out-of-scope material as referenced by highlighted comments was removed. Notably, we appreciate the updated pain management reference on acute pancreatitis. The authors extensively thank the reviewer for advice to remove hampering material discussing chronic pancreatitis. We feel the comments largely helped narrow the focus of the revised paper. We profusely thank the reviewer for the suggested edits and the time spent on comments.

Round 2

Reviewer 2 Report

Thank the author for carefully answering my question. I still believe that this article is a review of the acute pancreatitis pain syndrome, but the anatomy and physiology of the pancreas occupy a large space. Although these contents are meaningful for the pain of pancreatitis, they have a lot of content and affect the theme. However, this article has a novel theme and clinical significance, and I believe it has publication value.

Reviewer 3 Report

The authors addressed the comments and improved the quality of their manuscript.

Reviewer 4 Report

The review process resulted in a version that in my opinion can be accepted